# Predictors of Condom Use among College Students

**DOI:** 10.3390/ijerph21040433

**Published:** 2024-04-03

**Authors:** Maria José de Oliveira Santos, Elisabete Maria Soares Ferreira, Manuela Conceição Ferreira

**Affiliations:** 1Health Sciences Research Unit: UICISA: E-EsenfC/ESSIPV, Health School, University of Trás-os-Montes and Alto Douro, 5000-801 Vila Real, Portugal; 2Faculty of Psychology and Educational Sciences, University of Porto, 4200-135 Porto, Portugal; elisabete@fpce.up.pt; 3Health Sciences Research Unit: UICISA: E-EsenfC/ESSIPV, Health School of Viseu, Institute Polytechnic de Viseu, 3504-510 Viseu, Portugal; mmcferreira@gmail.com

**Keywords:** college students, condom, health promotion, nursing care, risk taking, sexual behavior

## Abstract

Consistent condom use is recognized as one of the most effective strategies to prevent unwanted pregnancies and sexually transmitted infections. Despite their effectiveness, condoms remain fairly well used among younger people. The conception of appropriate measures to change behaviors needs a deep understanding of the factors underlying poor adherence to condom use. This study aims to identify the predictors of condom use among college students. A cross-sectional, correlational, and predictive study was conducted involving a convenience sample of 1946 university students, with an average age of 21 years (20.74 ± 2.32). Pender’s Health Promotion Model (HPM) was used as a conceptual and methodological framework to understand the relationship between the predictors of condom use. An explanatory theoretical model of condom use behavior was established using path analysis. Condom use among young people is infrequent, with only 39.4% of respondents reporting consistent use. Perceived benefits, positive feelings, and interpersonal influences emerged as variables with the most explicitly positive influence on the commitment to condom use, a trend confirmed for both sexes. Commitment was the strongest predictor of condom use behavior (β = 0.580; *p* < 0.001). Pender’s HPM is effective in explaining the relationships between the predictors of condom use.

## 1. Introduction

Unprotected sex is, in some areas of the world, a significant factor contributing to morbidity and mortality due to sexually transmitted infections (STIs), including HIV/AIDS. The male condom is the most recommended method to prevent STIs, due to its effectiveness, ease of use, general availability, and low cost. According to the “World Health Organization (WHO)”, the prevalence and incidence of STIs remain high, even for the three curable STIs, chlamydia, gonorrhea, and syphilis, representing more than 2.5 million cases reported in 2021 [1]. Consistent condom use can reduce the risk of HIV/AIDS transmission, from an infected partner, by 80% to 90% [2]. Female condoms may be more effective than male condoms in preventing the transmission of STIs, but their effectiveness is difficult to quantify due to their low use rates [3]. Mitigating the impact of STIs is crucial for public health, and one highly recommended method is the consistent use of male condoms. The male condom is also a very suitable contraceptive method for adolescents and young adults. Promoting the correct and consistent use of condoms is a significant public health measure to improve sexual and reproductive health (SRH) among young individuals [4]. Several studies have revealed that condom use among higher-education students is inconsistent [5,6,7,8,9,10]. In a Portuguese study on university students’ health behaviors, more than half of the participants reported using condoms as a contraceptive method (73.7%), but 31% revealed that they used them inconsistently [11]. Incorrect condom use might compromise its effectiveness. Approximately one-third of heterosexual men report usage failures (31.3%) related to early rupture, delayed application, or premature removal, among other aspects [11,12,13]. In a North American study, 7% of women aged 15–44 who had used a condom in the past four weeks reported condom breakage or slipping off during intercourse or penis removal, contributing to the 25.8% that indicated condom use only partially during intercourse [14].

Some researchers sought to construct an explanatory model of the motivations behind (in)consistent condom use. These diverse motivations might be related to individual, cognitive, affective, psychosocial, and cultural factors [15,16]. The male gender, a more hedonistic and impulsive personality, and beliefs associated with reduced sexual pleasure are identified as the main barriers to effective condom use [17]. Cognitive and psychosocial variables, such as sexual health literacy, a more positive attitude, self-efficacy, and individual commitment to using condoms [18,19], the type of sexual partner, relationship status, and the ability to negotiate condom use with a sexual partner are typically associated with the adoption of the health-promoting behavior of consistent condom use [8,20,21]. Interpersonal influences commonly mediate these predictors, reflecting the importance attributed to significant persons, such as friends, romantic partners, family, and health professionals [19,22].

Situational influences arise from opportunities to create a favorable health promotion environment or may foster compromising behaviors. In an academic setting, characterized by a solid social component, eventually tied to alcohol or drug consumption, the conditions might be favorable to engage in risky sexual behaviors [23,24]. Psychoactive substances can negatively affect the ability to use and negotiate condom use or may even impair the assessment of a sexual partner’s potential risks [8,25,26,27]. This risk can increase due to socio-cultural factors, such as hookups with different partners each night, deemed as a way to have sex without the need to invest emotionally in a relationship [28]. Mobile phone applications like Tinder, which has about 10 million daily users, have boosted these one-night stand behaviors. Tinder’s young users (18–30 years old) have unveiled its use mainly for casual sex [29].

Between 30% and 60% of university students report at least one casual sex experience without condoms during their academic life [30,31,32]. The academic context is a situational influence that creates opportunities for risky behavior. Conversely, access to healthcare services, including STI prevention and contraception counselling, as well as free access to condoms, are potentially protective factors that encourage adherence to health-promoting behaviors [5,8]. The biggest challenge for public health policy is to provide an effective approach for young adults that can promote healthy sexuality, while also preventing adverse outcomes, such as STIs and unwanted pregnancy [33].

The Health Promotion Model (HPM), developed by Pender [34], has served as the theoretical and methodological framework for identifying the predictors of condom use. The author provides a guiding structure to explore the complex biological and psychological processes that motivate individuals to adopt health-promoting behaviors. It focuses on nursing intervention and follows different behavioral models that seek to understand changes in health-related behaviors. The model considers the multi-dimensional nature of interactions between actions taken to promote and maintain health.

The components of the model [34,35] include: (1) Individual characteristics and experiences that encompass the health-compromising behavior that needs to be changed and personal factors that influence adherence to health-promoting behaviors (biological, psychological, and sociocultural). (2) Behavior-specific cognition and affection, involving: (i) the perceived benefits of action and perceived barriers to action, which are positive or negative mental representations that strengthen or hinder the decision to adopt the behavior; (ii) perceived self-efficacy, reflecting the individual’s capacity to adopt healthier behaviors; (iii) subjective positive or negative feelings related to the health behavior; (iv) interpersonal influences, including norms and social support, and situational influences that can either facilitate or hinder specific health behaviors. (3) Behavioral outcomes that enable individuals to maintain the desired behavior, considering the plan of action commitment. Nursing interventions reinforce behavior commitment, support the weighing up of competing demands and preferences, and allow individuals to implement health-promoting behaviors into their lifestyle, resulting in significant health improvements.

This multifaceted approach enables us to identify modifiable factors crucial for developing targeted intervention programs, a pivotal aspect in safeguarding public health. This study aims to unravel the predictors of condom use among university students to formulate effective intervention strategies tailored to this population.

## 2. Materials and Methods

### 2.1. Study Design and Sample

We conducted a cross-sectional, descriptive, correlational study with a sample of 1946 students, aged from 18 to 29 years, from a university located in northern Portugal. In the present study, only participants who reported having sexual intercourse in the last year (*n* = 1496) were considered, as we intended to minimize memory bias associated with condom use.

### 2.2. Procedure for Data Collection

The technique used for sample selection was a random sampling of cluster groups. Classes (sample units) were grouped based on their representation of the scientific areas of study (Life Sciences, Health, and Human and Social Sciences) and the different years of study, and the classes were randomly selected from each group. In order to recruit students for the study, we directly solicited participation in the classroom, and participation was voluntary.

Data collection occurred after the university’s governing bodies and ethical committee approved the research protocol. The questionnaire was administered in the classroom, at the end of the class, and applied to all the students who volunteered to participate. The students were previously informed about the study’s objectives and its voluntary and anonymous nature. We provided all the students with details on completing the research protocol. The average completion time was 45 min. To ensure privacy and confidentiality, a sealed box was placed in the classroom into which the completed questionnaires were placed by the students themselves.

### 2.3. Measures

The research instrument used was a self-administered questionnaire designed to collect data on: (1) individual characteristics and experiences, including biological and sociocultural factors (gender, age, nationality, origin, education, parental income, area of study, and the importance of religion), experiences related to sexual and reproductive behavior (sexual activity, age at first instance of sexual intercourse and the contraceptive methods used, recent sexual activity, sexual activity in the context of a romantic relationship, contraception usage, and the specific contraceptive method used), and risky sexual behaviors (inconsistent condom use in the past year, having casual sexual partners, alcohol and drug consumption associated with sex); (2) behavior-specific cognition and affection (consistent condom use), which were assessed based on the variables proposed by Pender’s HPM (Table 1); (3) the health-promoting behavior to be implemented, which in this study was conceptualized as the consistent use of condoms.

### 2.4. Ethical Considerations

The study adhered to ethical principles inherent to research studies, respecting issues concerning the participants’ anonymity and confidentiality. The study obtained authorization from the university’s ethics committee (protocol No. 02/2016).

### 2.5. Data Analysis

The data were analyzed using SPSS 24.0 software (SPSS, Inc., Chicago, IL, USA). The association between individual characteristics and condom use was determined using the Chi-square test. The relationship between Pender’s HPM constructs and consistent condom use was established using Student’s *t*-test.

Only participants who reported having sexual intercourse in the last year (*n* = 1496) were included in the study. Before performing the statistical analysis, the variables related to condom use were adjusted with a control question to determine whether the student used condoms consistently. Consistent condom use was confirmed when the student indicated “always” in response to the control question. Frequent or sporadic use was considered inconsistent condom use and treated as non-use. Mediation models using structural equations were used to comprehend the relationships and interactions between the variables, from the hypothetical, theoretical model, and health-promoting behavior mediated by a commitment to action. Path analysis (AMOS, version 23) was also used to test the explanatory theoretical model of condom use behavior. The general model was adjusted for sex, the main confounding variable. This statistical technique allowed the examination of relationships and interactions between the variables belonging to the HPM and encompassed all the variables within a single regression model. Collinearity among the independent variables included in the model was assessed using variance inflation factors (VIFs). The absence of collinearity was accepted when the values were below five [36]. A significance level of *p* < 0.05 was considered in regard to all inferential statistics.

## 3. Results

The study was conducted using a sample of 1496 students (61.7% female and 38.3% male), with an average age of 21 years (20.92 ± 2.40), enrolled in different scientific areas at a university. Most participants were Portuguese (97.7%), and single (97.1%). They mainly came from low-income families (54.5% ≤ 2 minimum wages), with low levels of education (52.9% of mothers and 60.5% of fathers had only four years of school education), and undifferentiated jobs. The data also showed that the average age of sexual initiation was 17 years (17.0 ± 1.81). Only 17.3% of students reported that their first sexual experience occurred before they were 15 years old. The majority (96.4%) reported using contraception. The most used contraceptive method among female students was hormonal forms (52.9%), and by males was condoms (31.1%). The percentage of students reporting having had an STI was low (1.8%), as was the rate of unwanted pregnancies (3.1%). A substantial percentage of students reported having engaged in sexual risk behaviors in the last year, including inconsistent condom use (60.6%), casual partners (32.0%), and sexual intercourse associated with alcohol (33.0%) or drug (9.7%) consumption.

The influence of individual characteristics on the adoption health-promoting behaviors is presented in Table 2 for all participants and by gender, separately. The rate of inconsistent condom use is around 60% for both genders. The age of the student influenced (*p* < 0.05) consistent condom use, regardless of gender. It was observed that older students use condoms less frequently. Among younger students (≤19), consistent condom use was reported by about half of the respondents. This proportion decreases to less than one-third among students aged 25 or older. Female students attending Life Sciences and Healthcare study areas tend to adopt health-promoting behaviors more frequently (*p* = 0.027) than those studying in other fields. Among male students, this influence was not observed. Family income did not (*p* > 0.05) influence behavior adoption. The parental education level was redundant, along with the family’s income, with higher income families related to more highly educated parents.

The importance attributed to religion was found to influence behavior adoption. Religious believers have higher rates of consistent condom use (*p* = 0.011). The previous behavior, “condom use at first sexual intercourse”, used to assess the early concern for safe sex, significantly influenced (*p* = 0.001) the current behavior of male students. The tendency was similar among female students, but it was not significant (*p* = 0.176). Sex in a romantic relationship resulted in reduced condom use, both for female (*p* = 0.038) and male students (*p* < 0.001). Since older age and involvement in a romantic relationship were both factors that determine the abandonment of condoms, it was supposed that the same phenomenon applied, with older students being those involved in stable relationships. However, these two variables had no significant relationship (Ӽ^2^ = 0.009; *p* = 0.995).

The relationship between the variables related to condom use cognition and affection was evaluated by comparing the mean value of each variable, separately, according to gender and for all participants (Table 3). It was observed that most of the variables studied had a significant association with adopting health-promoting behaviors. Perceived benefits of action, positive feelings, and interpersonal influences were the variables with a more apparent positive influence on behavior, with a similar trend in both genders. The perception of the benefits of action and positive feelings regarding health-promoting behavior resulted in high scores on the scale, in both cases, for those who use condoms and those who do not, indicating that these aspects are already reasonably internalized by students. Despite the high scores observed, no statistical differences existed between students who adopted health-promoting behaviors and those who did not.

The interpersonal influences results indicate that students scored close to the centre of the scale, suggesting that the students do not particularly value this aspect. Perceived self-efficacy for condom use and situational influences contributed to engaging in health-promoting behaviors, albeit with a slightly different pattern for females and males. Self-efficacy for condom use was only significant for female students, while situational influences were significant for male students. Perceived barriers to action and negative feelings were the two variables that were negatively associated with health-promoting behaviors.

Perceived barriers represent a critical aspect of this behavior, as the mean scores were between 15 and 16 points, with a maximum score of 21. In the conceptualization of the determinants of health-promoting behaviors in the HPM, relationships and interrelationships were established between variables in the domains of cognition and affection, interpersonal influences, situational influences, and commitment to behavior (referred to as commitment to a plan of action in the HPM), and the behavior itself. Although univariate analysis allowed us to understand most of the relationships between the HPM constructs and condom use, it does not provide a comprehensive view of their combined effects. Modelling complex phenomena through structural equations is a technique based on multiple regressions that is suitable for quantifying the relative importance of these relationships, mainly because it allows an understanding of the role of the mediator. In the HPM, this role is assigned to the commitment to action variable, or, in the context of this study, the commitment to condom use.

Considering the differences observed in the univariate analysis between genders, this analysis was also conducted separately for female (Table 4, Figure 1a) and male students (Table 4, Figure 1b), as well for the entire sample (Table 4, Figure 2). It was observed that perceived benefits and barriers, as well as feelings, strongly influenced the mediator variable in both genders and, consequently, on the total sample studied, even though the values of the trajectory coefficients (β) were slightly different.

For female students, the perceived benefits of action were more critical (β = 0.239; *p* < 0.001) than for males (β = 0.129; *p* < 0.001), as were negative feelings. Conversely, the commitment to health-promoting behavior among male students was influenced more by positive feelings (β = 0.215; *p* < 0.001) than among female students (β = 0.100; *p* < 0.001).

Interpersonal influences may cause these trends in the entire sample. This HPM construct best predicts commitment to behavior for both genders, with a trajectory coefficient of 0.35. Situational influences did not prove to be significant predictors of commitment. In the male student’s sub-sample, this variable was not significant. It was excluded from the model, while in the female one and the total sample, although significant, the coefficient was low (β = −0.060 vs. β = −0.051, respectively). In line with the HPM, the trajectory coefficients between commitment to behavior and health-promoting behavior were statistically significant (*p* < 0.001). These associations were stronger for female (β = 0.643; *p* < 0.001) than for male students (β = 0.516; *p* < 0.001), indicating that female students are more effective in translating commitment into effective behavior. On the other hand, for male students (Figure 2), the only variable determining behavior adoption was being committed to it. No statistically significant direct trajectories were observed between HPM constructs and health-promoting behaviors. A small negative contribution (β = −0.078; *p* = 0.007) from interpersonal influences was added to commitment in female students. When we analyzed the entire population, the commitment variable showed the highest predictive capacity, explaining 58% of the variability in condom use behavior (β = 0.580; *p* < 0.001). There were also small direct contributions from perceived barriers (β = −0.045; *p* = 0.035), positive feelings (β = 0.065; *p* = 0.002), and interpersonal influences (β = −0.054; *p* = 0.023).

## 4. Discussion

University life corresponds to a phase of psychological and biological development for individuals. This highly appealing social context can result in intercourse, whether or not it occurs within a romantic relationship [6]. The sexual relationships in which these students engage, if not adequately protected, might represent an STI risk [37].

In light of current knowledge, the alternative to the unlikely option of sexual abstinence, consistent and correct condom use, is the most effective way to prevent STIs [1,4,15,37]. Thus, consistent condom use should be considered a health-promoting behavior [1,4,33,38,39,40]. Despite all the advantages associated with correct condom use, in this study, we observed low adherence, as less than 40% of students reported using condoms consistently. Several studies, both in Europe and North America, have also observed this trend, finding that the percentage of condom use among young people did not exceed 60% [1,4]. The results of a study involving university students revealed that the situation in Portugal is also concerning, with only 60.5% of students reporting the use of condoms consistently during sexual intercourse in the past 12 months [41]. The non-use or inconsistent use of condoms is particularly concerning, not only because of the short and long-term effects on reproductive health, but also because this is a period of health behavior consolidation that will be carried into their adult lives [42,43].

In the present study, we observed that students using condoms less frequently are older, are not religious, engage in sexual activities within the context of romantic relationships, and are male and female students studying Social Sciences. These trends are consistent with those previously observed in other studies [2,4,44,45]. Older students were less likely to report condom use and more likely to have sex with a casual partner than younger students. This trend may be associated with less integration into social life and the possible sexual and psychological immaturity of some younger students [18]. Another possible explanation for our findings is that only younger students received sexual education in high school. A law change was introduced in Portugal after 2009, which led to mandatory sex education, resulting in younger students receiving sex education in school, contrary to older ones. The level of religiosity can be used as a measure of attitudes toward sex-related activities, such as casual sex or the use of condoms, to avoid unplanned pregnancies or STIs [45]. Although, there is no conclusive evidence on the relationship between religiosity and condom use, some studies have found a positive association [45], another found a negative association [4], or no association [46,47]. In this regard, youth sexual and reproductive health promotion efforts should focus on providing comprehensive risk reduction sexual education regardless of religious background or beliefs, to lessen the likelihood of negative health outcomes [45]. 

A systematic review indicates that people use condoms more often during sexual encounters with casual partners than with stable partners. Additionally, people with multiple partners used condoms more frequently [48]. The denial of sexual risk in a romantic relationship is accompanied by the belief in affection, loyalty, and mutual trust as a guarantee of protection. Establishing trust is so essential that it precludes discussions on condom use for fear that broaching the topic would imply infidelity. Within romantic relationships, suggesting using condoms would be seen as indicating a lack of respect. Instead, it is expected that partners trust each other to protect themselves against STIs [49].

The results of this study show the coherence and suitability of the HPM in explaining the complexity of this problem, as most of the associations and interrelationships predicted by the authors are confirmed in our population sample. Although the main trends were similar, slight differences were observed between female and male students. Students generally perceive the benefits of condom use [4]. This perception, mediated by a commitment to action, is the best predictor of their practical use in the present study. Students acknowledge the benefits of condom use in preventing STIs. However, the fact that they choose not to use them is influenced by a groundless perception that they are not at risk [4,5,34]. The majority of students know that using condoms lowers the risk of STIs, but their main concern about unprotected sex is still pregnancy, as found in other studies [33,50]. In a recent study carried out in Portugal and Spain, the most frequently mentioned reasons for using condoms were the awareness of risk in specific situations (32%), concern about health including STIs (28%), behavioral control (14%), and concerns about health related to pregnancy (10%) [51].

Positive feelings related to the gratification from making a responsible choice were also identified as one of the best predictors of a commitment to condom use, particularly among male students, which contradicts the high rates of non-use observed. Perceived barriers likely influence the discrepancy, as this construct significantly contributes to the lack of commitment to condom use. Embarrassment associated with purchasing and using condoms, as explored in the perceived barriers, remains one of the factors that most significantly hinders consistent condom use [52,53].

Interpersonal influences have the most significant contribution to commitment to behavior, indicating that the opinions of significant people, along with those of their sexual partner, contribute to the process that determines the commitment to condom use [15,19,22]. These influences are significant for female students. In addition to the relationship between interpersonal influences and behavior through the commitment mediating variable, they also have a direct predictive capacity. The hedonistic and risk perspectives associated with engaging in sexual relations with casual partners or under the influence of alcohol or drugs, as considered in the situational influences, were expected to influence non-commitment. However, the results showed an impact with small trajectory coefficients, indicating their limited contribution. This trend might be underestimated in the present study, as these situational influences often occur in impulsive moments [54], which the students might not recognize.

The use of a purely rational approach to preventing diseases and reducing sexual risks is not very appealing to young adults who are healthy and influenced by powerful stimuli. Therefore, the decision to use a condom during sexual activity depends more on factors related to sexuality as a whole, including psychological, relational, and cultural aspects, as well as affective and situational conditions that characterize it. These factors are more significant than considerations related to preventive health. It is imperative that public health services prioritize the social promotion of condoms as a matter of citizenship. Condoms should be considered fundamental to respecting our health and that of others.

This study presents some limitations. It was conducted at a single university, using a non-probabilistic convenience sample, which may limit the generalization of the results. However, this limitation may have been mitigated by students coming from various Portuguese regions and the random selection of classes for data collection. The measures of sexual behavior were self-reported, which may not provide a precise view of and the actual behaviors of the participants due to potential bias related to social desirability, given the sensitive nature of the subject. Additionally, there may have been memory bias caused by temporal distancing.

## 5. Conclusions

Consistent condom use is a health-promoting behavior that should be encouraged among students. Therefore, by using Pender’s HPM as a theoretical framework, this study provided insights into understanding the determinants that have the most significant impact on the commitment to condom use and actual use. It was observed that most of the variables studied showed a significant association with a commitment to adopting health-promoting behaviors. The perceived benefits of the action, positive feelings, and interpersonal influences were the variables that exerted the most evident positive influence on the commitment to condom use, and this trend was observed in both female and male students. The perception of barriers to action and negative feelings negatively influenced the adoption of condom use behavior.

Commitment to condom use was the best predictor for the actual behavior. It mediates the influence of perceived benefits, positive feelings, and interpersonal influences on condom use behavior. The self-efficacy for condom use was generally high in the sample, and tendentially similar between groups, which might justify its low predictive value for condom use behavior, both directly, or through mediating the commitment to use. These findings highlight the importance of focusing on certain primary domains, like perceived benefits, positive feelings, and interpersonal influences, when working with university students to promote their commitment to and adoption of health-promoting behaviors, especially condom use consistency. By strategically addressing these determinants, public health initiatives can effectively promote and support the adoption of health-promoting behaviors, contributing significantly to the overall well-being and sexual health of university students.

## Figures and Tables

**Figure 1 ijerph-21-00433-f001:**
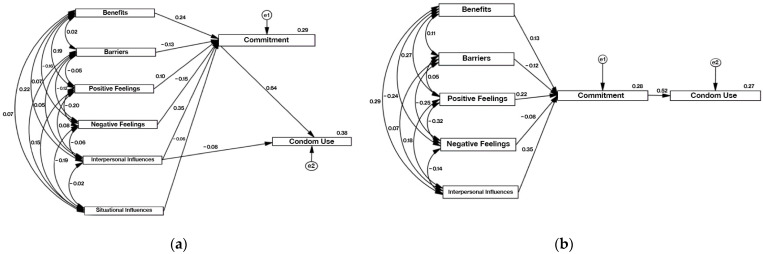
(**a**) This final refined model describes the association between the adoption of condom use as a health-promoting behavior, commitment to a plan of action, and the HPM constructs for female students; (**b**) final refined model describing the association between the adoption of condom use as a health-promoting behavior, commitment to a plan of action, and the HPM constructs for male students.

**Figure 2 ijerph-21-00433-f002:**
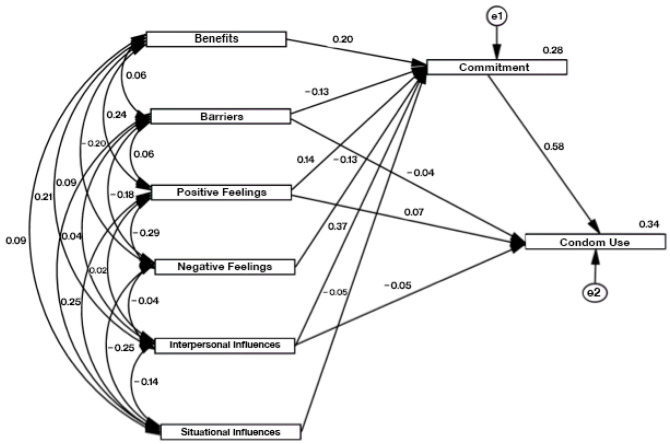
A refined model of the association between the adoption of the health-promoting behaviors of using condoms, commitment to a plan of action, and the HPM constructs for the entire population studied.

**Table 1 ijerph-21-00433-t001:** Predictive variables for condom use, according to Pender’s HPM.

Variables	Questions	Internal Consistency Cronbach’s Alpha	Score Range
Perceived benefits *	“By adopting preventive behaviors, I prevent future reproductive health complications.” “For me, it is easy to use condoms in my daily life.”	A = 0.442	Scores between 2 and 14
Negative feelings *	“Getting condoms from the health centre is an embarrassing situation.” “Talking to healthcare professionals about contraceptive use-related issues can be embarrassing.” “Buying condoms is embarrassing because it exposes my privacy.”	A = 0.720	Scores between 3 and 21
Positive feelings *	“Using and discussing contraceptive methods is part of responsible sexuality.” “I feel better about myself when I use contraceptive methods.”	A = 0.420	Scores between 2 and 14
Self-efficacy for condom use **	Self-efficacy for condom use, the Portuguese version of the Condom Use Self-Efficacy Scale (CUSES), consisting of 15 questions organized according to four factors (mechanisms, partner disapproval, assertiveness, intoxicants).	α = 0.820	Scores between 0 and 60
Interpersonal influences *	“People who are important to me advise me always to have and use condoms.” “It is important for sexual partners to talk about condom use.”	α = 0.594	Scores between 2 and 14
Situational influences *	“It is fun to have sexual experiences with casual partners.” “A good way to obtain sexual pleasure is to have sex under the influence of alcohol or drugs.”	α = 0.421	Scores between 2 and 14
Commitment to the plan of action *	“There is a high probability that I will use condoms over the next month.” “If I have sexual intercourse in the next month, I intend always to use condoms.”	α = 0.876	Scores between 2 and 14

* Questions were assessed using a Likert-type scale with seven response options (1—“completely disagree” to 7—“completely agree”). ** Questions were assessed using a Likert-type scale with five response options (0—“completely disagree” to 4—“completely agree”).

**Table 2 ijerph-21-00433-t002:** Influence of individual characteristics and experiences on health-promoting behaviors related to condom use.

Individual Characteristics and Experiences	Female Students		Male Students		Total	
	Use	Don’t use	Use	Don’t use	Use	Don’t use
n (%)	369 (39.9)	556 (60.1)	221 (38.7)	350 (61.3)	590 (39.4)	906 (60.6)
Age						
≤19	50.5	49.5	50.7	49.3	50.6	49.4
20–24	36.0	64.0	35.7	64.3	35.9	64.1
≥25	26.9	73.1	32.0	69.0	29.3	70.7
Ӽ^2^ (*p*)	20.535 (<0.001)		11.623 (0.003)		32.116 (<0.001)	
Field of Study						
Life Sciences and Healthcare	42.32	57.8	38.6	61.4	40.9	59.1
Human, Social, and Technology Sciences	35.4	64.6	39.1	60.9	37.0	63.0
Ӽ^2^ (*p*)	3.962 (0.027)		0.014 (0.906)		2.234 (0.135)	
Family Income						
<2 minimum wage	40.4	59.6	40.6	59.4	40.5	59.5
2–4 minimum wage	37.9	62.1	36.5	63.5	37.3	62.7
>4 minimum wage	42.2	57.4	38.1	61.9	40.2	59.8
Ӽ^2^ (*p*)	0.756 (0.685)		0.842 (0.656)		1.246 (0.536)	
Importance of Religion						
Limited/none	32.5	67.5	27.3	72.7	29.9	70.1
Moderate	37.3	62.7	40.0	60.0	38.4	61.6
High	43.2	56.8	41.3	58.7	42.5	57.5
Ӽ^2^ (*p*)	4.956 (0.084)		5.066 (0.079)		9.026 (0.011)	
Condom Use at First Sexual Intercourse						
Yes	41.3	58.7	41.7	58.3	41.5	58.5
No	36.5	63.5	24.2	75.8	33.2	66.8
Ӽ^2^ (*p*)	1.827 (0.176)		10.226 (0.001)		7.748 (0.003)	
Sexual Intercourse Within a Romantic Relationship						
Yes	39.1	60.9	34.9	65.1	37.7	62.3
No	54.2	45.8	54.0	46.0	54.0	46.0
Ӽ^2^ (*p*)	4.302 (0.038)		13.861 (<0.001)		16.099 (<0.001)	

**Table 3 ijerph-21-00433-t003:** Influence of HPM constructs related to commitment to a plan of action and health-promoting behaviors related to condom use.

HPM Constructs	Female Students		Male Students		Total	
	Use	Don’t Use	Use	Don’t Use	Use	Do Not Use
Perceived Benefits of Action Scale range: 2–14	12.56 ± 2.27	11.46 ± 2.50	11.99 ± 2.36	11.09 ± 26.0	12.35 ± 2.32	11.32 ± 2.53
Student’s t-test (*p*)	−6.861 (<0.001)		−−4.147 (<0.001)		−8.008 (<0.001)	
Perceived Barriers to Action Scale range: 3–21	15.51 ± 4.62	16.33 ± 4.51	15.01 ± 4.40	15.78 ± 4.44	15.32 ± 4.54	16.11 ± 4.48
Student’s t-test (*p*)	2.675 (0.008)		2.023 (0.044)		3.327 (0.001)	
Self-efficacy for Condom Use Scale range: 0–60	50.14 ± 7.56	48.93 ± 8.59	48.13 ± 9.47	47.86 ± 9.65	49.38 ± 8.38	48.52 ± 9.02
Student’s t-test (*p*)	−2.248 (0.025)		−0.329 (0.742)		−1.895 (0.058)	
Positive Feelings Scale range: 2–14	13.26 ± 1.59	12.65 ± 1.85	12.09 ± 2.05	11.03 ± 2.24	12.82 ± 1.86	12.03 ± 2.15
Student’s t-test (*p*)	−5.370 (<0.001)		−5.699 (<0.001)		36.501 (<0.001)	
Negative Feelings Scale range: 2–14	2.90 ± 1.88	3.44 ± 2.09	3.52 ± 2.27	4.16 ± 2.53	3.13 ± 2.06	3.72 ± 2.29
Student’s t-test (*p*)	4.023 (<0.001)		3.090 (0.002)		5.088 (<0.001)	
Interpersonal Influences Scale range: 2–14	8.84 ± 3.42	7.46 ± 3.57	10.91 ± 2.64	9.49 ± 3.21	9.61 ± 3.31	8.25 ± 3.57
Student’s t-test (*p*)	−5.789 (<0.001)		−5.669 (<0.001)		−7.460 (<0.001)	
Situational Influences Scale range: 2–14	11.95 ± 2.43	11.84 ± 2.64	9.09 ± 2.90	8.70 ± 3.06	10.87 ± 2.96	10.63 ± 3.20
Student’s t-test (*p*)	−0.595 (0.552)		−1.528 (0.0127)		−1.517 (0.130)	

**Table 4 ijerph-21-00433-t004:** Coefficients in the refined, final model to predict the adoption of health-promoting behaviors, namely using condoms (dependent variable), commitment to a plan of action (mediator variable), and HPM constructs (independent variables).

Commitment to the Desired Behavior	HPM Constructs	C.R.	*p*	β
Female Students				
Commitment to condom use	Perceived benefits	8.218	***	0.239
	Perceived barriers	−4.497	***	−0.126
	Positive feelings	3.518	***	0.100
	Negative feelings	−5.201	***	−0.151
	Interpersonal influences	12.458	***	0.354
	Situational influences	−2.159	0.031	−0.060
Health-promoting behavior	Interpersonal influences	−2.705	0.007	−0.078
	Commitment to behavior	22.433	***	0.643
Male Students				
Commitment to condom use	Perceived benefits	3.317	***	0.129
	Perceived barriers	−3.175	0.001	−0.120
	Positive feelings	5.568	***	0.215
	Negative feelings	−1.971	0.049	−0.078
	Interpersonal influences	9.321	***	0.350
Health-promoting behavior	Commitment to a plan of action	14.280	***	0.516
Total Sample				
Commitment to condom use	Perceived benefits	8.575	***	0.200
	Perceived barriers	−5.719	***	−0.130
	Positive feelings	5.825	***	0.136
	Negative feelings	−5.269	***	−0.126
	Interpersonal influences	16.263	0.001	0.370
	Situational influences	−2.238	0.25	−0.051
Health-promoting behavior	Commitment to behavior	24.031	***	0.580
	Perceived barriers to action	−2.065	0.039	−0.045
	Positive feelings	3.047	0.002	0.065
	Interpersonal influences	−2.271	0.023	−0.054

Notes: *** *p* < 0.001; values of VIF < 5 (range between 1.035 and 1.186) indicate no collinearity problems between the variables.

## Data Availability

The data that support the findings of this study are available upon reasonable request from the corresponding author. The data in this article are part of a doctoral thesis, whose data can be found at: https://repositorio-aberto.up.pt/handle/10216/105488 (accessed on 10 July 2017).

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
