# Peer review of "Predictors of Condom Use among College Students"

_ijerph, 2024, doi:10.3390/ijerph21040433_

Round 1
Reviewer 1 Report
Comments and Suggestions for Authors
I sugest including discussions about sexual orientation and gender identity, if it has not been analyzed, recognize these aspects as a gap.
Author Response
Reviewer 1
Comments 1- I suggest including discussions about sexual orientation and gender identity, if it has not been analyzed, recognize these aspects as a gap.
Response 1 - Thank you for your valuable suggestion. The number of self-reported non-heterosexual students were residual in the samples. It was studied, but with a very limited interest due to the very low representation of this group. We have acknowledged this limitation in the study's limitations section.
Reviewer 2 Report
Comments and Suggestions for Authors
This study aimed to investigate the associations of condom use among university students.
I have the following concerns:
1: The authors published a study using the same data in 2018 investigating the associations with sexual behavior, including condom use (PMID: 28617971). Then, they published a qualitative study using Pender's Health Promotion Model (PMID: 35920496) to assess perceptions of sexual behavior, including condom use. What is the merit of the current study?
2: Methods: More data about the sampling methods and recruitment should be provided.
3: Sample characterization should be moved to results.
4: The study population included 1946 students. However, 17.6% of them reported that they never had sex. They should have been excluded from the sample. The inclusion criteria should be restricted to sexually active students.
5: The analysis was conducted on 1496 students only. No justifications were given for this difference. Per a previous study by the authors, those are the students who reported having sex during the previous year.
6: One of the main disadvantages of observational study is the high possibility of confounders. Thus, conducting regression analysis and adjusting the results for potential confounders is very important. Yet, the authors only provided unadjusted results.
7: 95% CIs should be added.
8: Discussion: Several findings were not discussed. The results of the current study were not compared with those of previous national and international studies. In addition to these comparisons, the authors should discuss the results from a social perspective.
Comments on the Quality of English Language
Moderate editing is needed, especially with numbers.
Author Response
Reviewer 2
I have the following concerns:
Question R2 -1: The authors published a study using the same data in 2018 investigating the associations with sexual behavior, including condom use (PMID: 28617971). Then, they published a qualitative study using Pender's Health Promotion Model (PMID: 35920496) to assess perceptions of sexual behavior, including condom use. What is the merit of the current study?
Response 1: We appreciate your careful reading of the article and valuable suggestions, which will certainly contribute to improving its overall quality. There are many different factors that can impact how people behave, some of which are easy to observe while others are hidden. Structural equation models offer a way to capture the complex relationships between these variables, including both direct and indirect connections. In this study, structural equation modelling allowed us to analyse how the identified theoretical variables relate to each other and how they can predict condom use. Overall, these findings are likely to help researchers and health care professionals improving theoretical models predicting condom use and preventing the spread of sexually transmitted infections.
Question 2: Methods: More data about the sampling methods and recruitment should be provided.
Response 2: Thank you for your valuable suggestion. More information about sample selection and participant recruitment methods was included in the methodology chapter.
Comments 3: Sample characterization should be moved to results.
Response 3: As the reviewer suggested, the sample characterisation information was transferred to the results chapter.
Comments 4: The study population included 1946 students. However, 17.6% of them reported that they never had sex. They should have been excluded from the sample. The inclusion criteria should be restricted to sexually active students.
Comments 5: The analysis was conducted on 1496 students only. No justifications were given for this difference. Per a previous study by the authors, those are the students who reported having sex during the previous year.
Responses 4 and 5: We acknowledge the reviewer comments on this issue that might be confusing for the reader. This study was performed with 1946 students, but, in fact only 1496 had sexual intercourse in the last year. The writing and sample characterization were updated accordingly.
Comments 6: One of the main disadvantages of observational study is the high possibility of confounders. Thus, conducting regression analysis and adjusting the results for potential confounders is very important. Yet, the authors only provided unadjusted results.
Response 6: Thank for your pointing. Although it was not mentioned, the data were adjusted for sex, the main confounding variable. That information is now in the data analysis section
Comments 7: Discussion: Several findings were not discussed. The results of the current study were not compared with those of previous national and international studies. In addition to these comparisons, the authors should discuss the results from a social perspective.
Response 7 - Thank you for your valuable suggestion. The discussion chapter was improved, we included some articles that allowed us to compare our results with those of other studies. We were unable to compare HPM data as we did not find studies applying this model related to condom use.
Reviewer 3 Report
Comments and Suggestions for Authors
The paper entitled: “Predictors of Condom Use Among College Students” aims to unravel predictors of condom use among university students to formulate effective intervention strategies tailored to this population.
To identify the predictors of condom use, the authors used the Health Promotion Model developed by Pender. This model includes Individual characteristics and experiences, behaviour-specific cognitions and affection and behaviour outcomes.
The non-use of condoms is concerning because condom use is the most effective strategy to prevent unwanted pregnancies and sexually transmitted infections. The authors concluded that students using condoms less frequently were older, were not religious, and engaged in sexual activities within the context of romantic relationships. According to the authors, these trends were observed in other studies. However, these results were counterintuitive as older, non-religious and students involved in romantic relationships should be more committed to using condoms. The authors quote three studies concluding that older, not religious, and students involved in romantic relationships are less committed to using condoms. However, there are no other studies indicating that age, religion and having a romantic relationship behave differently; id est, that consistent with intuitive thought conclude that younger, religious, and engaged in sexual activities within the context of non-romantic relationships use condoms less frequently?
In the conclusion, the authors mention two ideas that look contradictory. On the other hand, the authors say: “Contrary to expected, condom use self-efficacy showed a weak prediction ability for both commitment and actual behaviour”. On the other hand, the authors state: “The results show a strong association between commitment and condom use”. The authors should resolve this apparent contradiction.
Finally, I am concerned about the representativity of the sample used. The authors used a quite large sample (1946 university students). However, it was a convenience sample, and students were interviewed in only one university located in northern Portugal. A much smaller probabilistic sample with 400 students would have resulted in more reliable data.
Author Response
Dear Reviewers
Thank you for your valuable suggestions and I will send you a response to your considerations.
Reviewer 1
Comments 1- I suggest including discussions about sexual orientation and gender identity, if it has not been analyzed, recognize these aspects as a gap.
Response 1 - Thank you for your valuable suggestion. The number of self-reported non-heterosexual students were residual in the samples. It was studied, but with a very limited interest due to the very low representation of this group. We have acknowledged this limitation in the study's limitations section.
_____________________________________________________________________________
Reviewer 2
I have the following concerns:
Question R2 -1: The authors published a study using the same data in 2018 investigating the associations with sexual behavior, including condom use (PMID: 28617971). Then, they published a qualitative study using Pender's Health Promotion Model (PMID: 35920496) to assess perceptions of sexual behavior, including condom use. What is the merit of the current study?
Response 1: We appreciate your careful reading of the article and valuable suggestions, which will certainly contribute to improving its overall quality. There are many different factors that can impact how people behave, some of which are easy to observe while others are hidden. Structural equation models offer a way to capture the complex relationships between these variables, including both direct and indirect connections. In this study, structural equation modelling allowed us to analyse how the identified theoretical variables relate to each other and how they can predict condom use. Overall, these findings are likely to help researchers and health care professionals improving theoretical models predicting condom use and preventing the spread of sexually transmitted infections.
Question 2: Methods: More data about the sampling methods and recruitment should be provided.
Response 2: Thank you for your valuable suggestion. More information about sample selection and participant recruitment methods was included in the methodology chapter.
Comments 3: Sample characterization should be moved to results.
Response 3: As the reviewer suggested, the sample characterisation information was transferred to the results chapter.
Comments 4: The study population included 1946 students. However, 17.6% of them reported that they never had sex. They should have been excluded from the sample. The inclusion criteria should be restricted to sexually active students.
Comments 5: The analysis was conducted on 1496 students only. No justifications were given for this difference. Per a previous study by the authors, those are the students who reported having sex during the previous year.
Responses 4 and 5: We acknowledge the reviewer comments on this issue that might be confusing for the reader. This study was performed with 1946 students, but, in fact only 1496 had sexual intercourse in the last year. The writing and sample characterization were updated accordingly.
Comments 6: One of the main disadvantages of observational study is the high possibility of confounders. Thus, conducting regression analysis and adjusting the results for potential confounders is very important. Yet, the authors only provided unadjusted results.
Response 6: Thank for your pointing. Although it was not mentioned, the data were adjusted for sex, the main confounding variable. That information is now in the data analysis section
Comments 7: Discussion: Several findings were not discussed. The results of the current study were not compared with those of previous national and international studies. In addition to these comparisons, the authors should discuss the results from a social perspective.
Response 7 - Thank you for your valuable suggestion. The discussion chapter was improved, we included some articles that allowed us to compare our results with those of other studies. We were unable to compare HPM data as we did not find studies applying this model related to condom use.
____________________________________________________________________________
Reviewer 3
Comments 1 - The non-use of condoms is concerning because condom use is the most effective strategy to prevent unwanted pregnancies and sexually transmitted infections. The authors concluded that students using condoms less frequently were older, were not religious, and engaged in sexual activities within the context of romantic relationships. According to the authors, these trends were observed in other studies. However, these results were counterintuitive as older, non-religious and students involved in romantic relationships should be more committed to using condoms. The authors quote three studies concluding that older, not religious, and students involved in romantic relationships are less committed to using condoms. However, there are no other studies indicating that age, religion and having a romantic relationship behave differently; id est, that consistent with intuitive thought conclude that younger, religious, and engaged in sexual activities within the context of non-romantic relationships use condoms less frequently?
Response 1: We acknowledge the reviewer comment. This segment of text was unclear, probably due to our intention to highlight the lack of interest of the self-efficacy to condom use in the model. We believe that this lack of interest is related to the high scores obtained in this construct for the generality of the students. Its uniform tendency was probably in the basis of its low predictive interest. In the actual writing of conclusions, we tried to make this idea more straightforward.
Comments 2 - In the conclusion, the authors mention two ideas that look contradictory. On the other hand, the authors say: “Contrary to expected, condom use self-efficacy showed a weak prediction ability for both commitment and actual behaviour”. On the other hand, the authors state: “The results show a strong association between commitment and condom use”. The authors should resolve this apparent contradiction.
Response 2 - We agree with the statement provided. The outcomes can be accounted for by the fact that senior students tend to engage in sexual activities within the context of a romantic relationship. According to various studies, being in a romantic relationship can result in less frequent use of condoms. In a monogamous relationship, individuals may believe that they are not at risk of contracting a sexually transmitted infection (STI) and may stop using condoms. Additionally, trust in a partner is a significant factor in the decision to give up condom use, as asking a partner to use a condom can sometimes be seen as a breach of trust. The role of religion in this decision is still uncertain, as there is little consensus on its impact due to diverse results.
Comments 3 - Finally, I am concerned about the representativity of the sample used. The authors used a quite large sample (1946 university students). However, it was a convenience sample, and students were interviewed in only one university located in northern Portugal. A much smaller probabilistic sample with 400 students would have resulted in more reliable data.
Response 3: We agree with the referee; a smaller sample would be enough. Therefore, we e followed this approach with more students viewing the study of eventual course year and area of studies effects. It should also be mentioned that this study is part of a broader investigation, with the purpose of the broader research being to build an intervention program to promote the sexual and reproductive health of students at this university.
Round 2
Reviewer 2 Report
Comments and Suggestions for Authors
No more comments
Comments on the Quality of English LanguageMinor editing is needed.